# Fatty Acid Desaturation Is Suppressed in Mir-26a/b Knockout Goat Mammary Epithelial Cells by Upregulating *INSIG1*

**DOI:** 10.3390/ijms241210028

**Published:** 2023-06-12

**Authors:** Lu Zhu, Hongyun Jiao, Wenchang Gao, Lian Huang, Chenbo Shi, Fuhong Zhang, Jiao Wu, Jun Luo

**Affiliations:** Shaanxi Key Laboratory of Molecular Biology for Agriculture, College of Animal Science and Technology, Northwest A&F University, Yangling, Xianyang 712100, China; zhulu@nwafu.edu.cn (L.Z.); j17760232361@163.com (H.J.); gaowenchang666@163.com (W.G.); kin8248806@163.com (L.H.); shichenbo@nwafu.edu.cn (C.S.); zhang_fuhong@nwafu.edu.cn (F.Z.); wujiao2019@nwafu.edu.cn (J.W.)

**Keywords:** miR-26a, miR-26b, lipogenesis, unsaturated fatty acid, *INSIG1*, CRISPR/Cas9, goat mammary epithelial cells

## Abstract

MicroRNA-26 (miR-26a and miR-26b) plays a critical role in lipid metabolism, but its endogenous regulatory mechanism in fatty acid metabolism is not clear in goat mammary epithelial cells (GMECs). GMECs with the simultaneous knockout of miR-26a and miR-26b were obtained using the CRISPR/Cas9 system with four sgRNAs. In knockout GMECs, the contents of triglyceride, cholesterol, lipid droplets, and unsaturated fatty acid (UFA) were significantly reduced, and the expression of genes related to fatty acid metabolism was decreased, but the expression level of miR-26 target insulin-induced gene 1 (*INSIG1*) was significantly increased. Interestingly, the content of UFA in miR-26a and miR-26b simultaneous knockout GMECs was significantly lower than that in wild-type GMECs and miR-26a- and miR-26b-alone knockout cells. After decreasing *INSIG1* expression in knockout cells, the contents of triglycerides, cholesterol, lipid droplets, and UFAs were restored, respectively. Our studies demonstrate that the knockout of miR-26a/b suppressed fatty acid desaturation by upregulating the target *INSIG1*. This provides reference methods and data for studying the functions of miRNA families and using miRNAs to regulate mammary fatty acid synthesis.

## 1. Introduction

Goat milk is rich in short-chain and medium-chain saturated fatty acids and unsaturated fatty acids, high in protein, and has low allergenicity for humans [1,2]. Furthermore, goat milk contains a high concentration of linoleic acid (C18:2) and other unsaturated fatty acids, which are likely to lower blood cholesterol and promote antioxidation [3,4]. The milk fat globules in goat milk are small, and the proportion of milk fat globules less than 5 μm is more than 80%, which is beneficial for their absorption and digestion in the human intestinal tract [5]. Therefore, research on the molecular mechanism of regulating the lipid content in goat milk has received extensive attention.

MiRNAs are a group of small non-coding RNAs that mediate RNA silencing. They are involved in most encoded transcriptional proteins and are involved in all developmental and pathological processes in animals [6,7,8]. Many studies have shown that the regulation of the biological function by multiple miRNAs or miRNA families is synergistic [9,10]. Three miRNAs of the miR-17/92 family synergistically target the phosphatase and tensin homolog (PTEN) to inhibit IL-12-mediated immune defense by regulating the PI3K-Akt-GSK3 pathway [11]. miR-204-5p and miR-211 synergistically target Casein Alpha S1 (CSN1S1) to inhibit αS1-casein content in goat mammary epithelial cells [12]. Meanwhile, miRNA affects the content of short-chain and unsaturated fatty acids in milk from goats [13]. miR-26 family expression levels are significantly interrelated with the fatty acid composition in goat milk [7]. Members of the goat miR-26 family include miR-26a and miR-26b, whereby their host genes are the encoding protein family of Carboxy-terminal domain RNA polymerase II polypeptide A small phosphatase (*CTDSP*). miR-26a and miR-26b can be found inside the intronic regions of CTD small phosphatase-like (*CTDSPL*) and CTD small phosphatase 1 (*CTDSP1*), respectively. Most miRNAs are co-expressed or regulated with their host genes [14,15]. The gene expression involved in lipogenesis was reduced after suppressing the expression of miR-26a/b in goat mammary epithelial cells (GMECs) [16]. Insulin-induced gene 1 (*INSIG1*) is one of the key regulators of maintaining intracellular lipid metabolism homeostasis by affecting the activation of sterol regulatory element-binding proteins (SREBPs) and regulating the expression of fatty acid desaturase 2 (*FADS2*), stearoyl-CoA desaturase (*SCD1*), and others [17]. Interestingly, it is speculated that miR-26a/b regulates triglyceride synthesis by targeting *INSIG1* [16]. However, the miR-26a/b endogenous regulatory mechanism in fatty acid metabolism is unclear in GMECs.

In this study, we obtained miR-26 knockout GMECs to verify their effect on mammary lipogenesis using the CRISPR/Cas9 system in vitro. Furthermore, the target gene *INSIG1* of miR-26 was knocked down in knockout cells to explore the molecular mechanisms of miR-26a and miR-26b involved in adipogenesis in GMECs. This provides the possibility to manipulate milk fat content in goat milk by regulating miRNA in the future.

## 2. Results

### 2.1. Knockout Cells Were Obtained Using the CRISPR/Cas9 System with Multiple sgRNAs

We constructed two dual sgRNA and Cas9 co-expression vectors and simultaneously transfected these two vectors into GMECs to achieve the co-expression of three or four sgRNAs and Cas9 in cells to knock out miR-26a or miR-26b alone and simultaneously. We found a 64 bp deletion in the pre-miR-26a sequences of miR-26a-9 and miR-26ab-12 cell lines miss 62 bp (Figure 1A and Appendix A). The pre-miR-26b sequences in miR-26b-1 and miR-26ab-12 cell lines have wild-type and deletion sequences (Figure 1B). Then, we detected the expression levels of miR-26a and miR-26b in cells. The expression of miR-26a-5p and miR-26a-3p (mature miRNAs of miR-26a) in miR-26a-9 (miR-26a KO) and miR-26ab-12 (miR-26 KO) cells were significantly downregulated, and the expression of miR-26b-5p and miR-26b-3p (mature miRNAs of miR-26b) in miR-26b-1 (miR-26b KO) and miR-26ab-12 (miR-26 KO) cells was also significantly decreased (Figure 2A).

The expression levels of host genes *CTDSPL* and *CTDSP1* were reduced in knockout GMECs (Appendix A). Additionally, we found that the expression of the miR-26a and miR-26b target gene *INSIG1* in GMECs was upregulated in knockout cells (Figure 2B,C). We simultaneously transferred multiple sgRNAs into the cells, which may have improved the success rate of gene editing, but may have led to more off-targets. Therefore, we further verified off-target sites using the T7EN1 restriction endonuclease assay, and only one off-target site was found (Appendix A).

### 2.2. Knockout of miR-26a/b Inhibits the Expression of Genes Involved in Lipid Metabolism

Furthermore, the expression levels of the lipid-metabolism-related genes were detected in knockout GMECs. After the knockout of miR-26a/b, the mRNA expression of lipid-metabolism-related transcription factors was significantly downregulated (Figure 3A). At the same time, the mRNA expression levels of genes for fatty acid synthesis in knockout GMECs were also significantly lower than those in wild-type GMECs (Figure 3B). The expression levels of fatty acid desaturase 2 (*FADS2*) and stearoyl-CoA desaturase (*SCD1*), which are related to fatty acid desaturation, were significantly downregulated in knockout GMECs, and the *SCD1* expression levels in knocked-out miR-26ab (miR-26a and miR-26b co-knocked out, miR-26ab KO) GMECs were significantly less than those in the miR-26a (miR-26a KO) or miR-26b (miR-26b KO) GMECs (Figure 3C). In addition, the knockout of miR-26a/b resulted in a decreased expression of triglyceride-related genes (Figure 3D).

### 2.3. Knockout of miR-26a/b Downregulated the Content of Lipid Droplets, Cholesterol, and TAG in GMECs

The lipid droplet content was significantly reduced in knockout GMECs (Figure 4A,B), and that in miR-26a KO cells was significantly lower than that in miR-26b KO cells. However, the content in miR-26ab KO was not less than that in the miR-26a KO group (Figure 4A,B). The cellular concentrations of cholesterol and triglycerides in the knockout GMECs were significantly suppressed (Figure 4C,D). The cholesterol concentrations in miR-26b KO and miR-26ab KO GMECs were significantly higher than those in miR-26a KO, but with no statistical difference between miR-26b KO and miR-26ab KO GMECs (Figure 4C). The simultaneous knockout of miR-26a and miR-26b did not enhance the effect of knockout miR-26a or knockout miR-26b on cellular triglyceride content (Figure 4D).

### 2.4. Double Knockout of miR-26a and miR-26b Enhanced the Inhibition of the UFA Percentage

Next, we analyzed the fatty acid composition in the knockout GMECs (Appendix A). The content of palmitic acid (C16:0) was significantly increased in the simultaneous knockout of miR-26a and miR-26b GMECs, but that in knockout miR-26a GMECs and knockout miR-26b GMECs did not significantly increase (Figure 5A). The percentages of oleic acid (C18:1) and linoleic acid (C18:2) in knockout miR-26ab GMECs were significantly suppressed (Figure 5B). The percentage of unsaturated fatty acids (UFAs) was significantly reduced in knockout GMECs (Figure 5C). Moreover, knocking out miR-26a and miR-26b at the same time enhanced the inhibitory effect of knocking out miR-26a or knocking out miR-26b on the proportion of unsaturated fatty acids (Figure 5C).

### 2.5. miR-26a/b Affects the Lipid Metabolism of Breast Cells by Targeting INSIG1

We previously found that *INSIG1* is the target gene of miR-26a and miR-26b, and deduced that miR-26a and miR-26b probably affect lipid metabolism through *INSIG1* in GMECs. Therefore, we performed a rescue assay by transfecting siINSIG1 in knockout GMECs to detect lipid metabolism. The mRNA and protein levels of *INSIG1* in knockout cells were downregulated by transfecting siINSIG1 (Figure 6A,B). After interfering with the expression of *INSIG1* in knockout GMEC, the cellular concentration of TAG and cholesterol were significantly increased (Figure 6C,D). Moreover, the lipid droplet content of the knockout GMECs with siINSIG1 was significantly higher than that in the control group (Figure 7A,B). After the expression of *INSIG1* in knockout GMECs was disturbed, the fatty acid compositions were also changed (Appendix A), and the proportion of UFA was significantly upregulated (Figure 7C–E).

## 3. Discussion

Goat milk is rich in fatty acids, which is the main reason for its unique healthcare function. Exploring the molecular mechanism of regulating goat milk fatty acid synthesis is an important basis for optimizing the composition and content of goat milk fatty acids [18]. Previous studies found that the miR-26 family promotes lipid metabolism in goat mammary epithelial cells, which may be mediated by the target *INSIG1* [16]. However, the role of the miR-26 family in the regulation of fatty acid metabolism in the mammary gland has not been explored in depth. In this study, we first obtained knockout miR-26 GMECs using the CRISPR/Cas9 method, and further validated the role of the miR-26 family in fatty acid metabolism.

The CRISPR/Cas9 system achieves targeted gene editing through sgRNA, but because the mature sequence of miRNA only has about 22 nucleic acids, the CRISPR/Cas9 of a single sgRNA has less of an effect on miRNA genome editing, while the efficiency of double sgRNA is higher [19]. Previously, we found that a single miRNA was knocked out in GMECs using the CRISPR/Cas9 system with double sgRNA [18,20,21]. MiR-26a and miR-26b knockout cell lines were obtained by transfecting two double-sgRNA-containing vectors into GMECs, using puromycin for screening and cell monoclonal culture. Eight cell lines were obtained, whereby four of which had altered pre-miR-26b sequences in the miR-26b knockout group (Appendix A). Furthermore, in the miR-26ab knockout group (double-knockout miR-26a and miR-26b), we obtained 13 cell lines. However, only one cell line had both the pre-miR-26a and pre-miR-26b sequences knocked out (Appendix A). The sequence of the off-target site is shown in Appendix A. Only one off-target site (B-OT-2-4) was found in miR-26b KO GMECs (Appendix A). This off-target site is located near 81,746,540 in goat chromosome 7 (Appendix A). Fortunately, this off-target site is located in the intergenic region and spaced 99 kb from the nearest coding region (Appendix A). Presumably, this off-target location has no regulatory effect on the cell. However, it is also possible that unannotated functional genes or non-coding regulatory sequences exist in the vicinity of off-target sites.

Interestingly, the expression of *CTDSPL* and *CTDSP1*, the host genes of miR-26a and miR-26b, was significantly reduced in knockout cells. This may be because the pre-miR-26a/b targeted by the sgRNA is located in the intronic region of *CTDSPL/1* [22,23]. The disruption of gene introns may affect splicing during gene transcription, which in turn affects the level of transcription [24]. However, the specific regulatory mechanism has not been explored. Previous studies have reported that miR-26a/b expressed the same trend as the host gene *CTDSPL/1* [7]. In addition, the expression of the miR-26b host gene *CTDSP1* was downregulated in miR-26a knockout cells, and the expression of the miR-26a host gene *CTDSPL* was also downregulated in miR-26b knockout cells. Liu confirmed that the overexpression of miR-26a in human breast cancer cells significantly increased the expression of the host genes *CTDSPL* and *CTDSP2*, suggesting a self-perpetuating loop of miR-26a [25]. It is more likely that miR-26a was knocked out, resulting in a decrease in *CTDSPL* expression, not editing the intron. However, in knockout cells, whether *CTDSPL* and *CTDSP2* are involved in the regulation of lipid metabolism needs to be further explored. At the same time, we found that the expression of the target gene *INSIG1* was significantly increased in knockout GMECs. However, the protein expression of *INSIG1* in GMECs with simultaneous knockouts of miR-26a and miR-26b did not change significantly compared to its expression in GMECs with single knockouts. The changes in the expression of host genes and target genes indirectly demonstrated that miR-26a and miR-26b were altered in knockout cells, respectively.

Many miRNAs have emerged as important regulators of lipid metabolism [26,27,28]. MiR-26a has been proven to regulate lipid metabolism in multiple organizations. The specific overexpression of miR-26a in adipose tissue modestly reduced visceral fat pad mass and lipid levels in mice [29]. In addition, insulin sensitivity was improved and hepatic glucose production and fatty acid synthesis were reduced in miR-26a-overexpressing mice fed a high-fat diet [30]. Ali found that miR-26a attenuated triglyceride accumulation via the repression of lipogenesis and induction of autophagy in vitro in human HepG2 cells [31]. ER stress, lipid accumulation, and hepatic steatosis are exacerbated in mice lacking miR-26a after high-fat diet induction [32]. High-fat-diet-induced obesity is attenuated in transgenic mice overexpressing miR-26, and miR-26 at least partially blocked adipogenesis by inhibiting F-Box and leucine-rich repeat protein 19 (*Fbxl19*) expression [33]. The above results suggest that miR-26a inhibits lipid accumulation in adipose tissue and liver tissue. This is the exact opposite of our result, finding that miR-26a promotes adipogenesis in breast tissue. The content of triglycerides, cholesterol, and lipid droplets in the knockout miR-26a cells was significantly lower than that in the control group. This may be because miRNA targets different genes that exert different roles in different tissues. The expression of miR-26b showed an upward trend during the differentiation of human preadipocytes, and the differentiation of preadipocytes was also inhibited when miR-26b was inhibited [34]. MiR-26b−5p promotes goat preadipocyte differentiation by targeting fibroblast growth factor 21 (*FGF21*), which was found after the transient overexpression of miR-26b−5p in cells [35]. An overexpression of miR-26b significantly accelerated the mRNA expression of peroxisome proliferator-activated receptor γ (*Pparγ*) and fatty acid synthase (*Fasn*) in mouse preadipocyte cell lines [36]. After transiently reducing the expression of miR-26a and miR-26b in human pluripotent adipose stem cells via the transfection of inhibitors, the lipid droplets and triglyceride content in the cells were inhibited, and the miR-26 family and its target gene *ADAM* metallopeptidase domain 17 (*ADAM17*) regulated human fat production and energy dissipation development [37].

Wang inhibited miR-26a and miR-26b expression in GMECs via RNAi and found that the downregulation of miR-26 suppressed the accumulation of lipid droplets in the cells [16]. In addition, the expression of miR-26 is directly related to milk fatty acid composition and underscores the significance of miRNAs in milk fat synthesis regulation [7]. We found that miR-26a and miR-26b had no additive effect on the content of lipid droplets, triglycerides, and cholesterol, plus the expression of related genes in GMECs. Interestingly, we found that after miR-26a and miR-26b were knocked out together, the unsaturated fatty acid content in GMECs was lower than that in the wild-type, knockout-miR-26a-alone and knockout-miR-26b-alone GMECs. In addition, the expression of the gene *SCD1*, which regulates the process of fatty acid desaturation, also showed the same trend. Therefore, the knockout of miR-26 by CRISPR/Cas9 can inhibit the proportion of unsaturated fatty acids in GMECs.

Our previous study found that miR-26a and miR-26b inhibit the expression of *INSIG1* in GMECs, and confirmed that *INSIG1* is the target gene of miR-26a and miR-26b in 293T cells. It is speculated that the miR-26 family may participate in lipid metabolism by regulating the target *INSIG1* [16]. Therefore, we propose the hypothesis that miR-26 may participate in the regulation of fatty acid metabolism by targeting *INSIG1*. Then, the siRNA of *INSIG1* was transfected into GMECs knocking out the miR-26 family for rescue experiments. The data showed that the contents of lipid droplets, triglycerides, cholesterol, and unsaturated fatty acids were upregulated after the expression of INSIG1 was inhibited in the knockout cells. *INSIG1* is involved in maintaining the homeostasis of intracellular lipid metabolism, mainly by controlling the transfer of SCAP-SREBP complexes from the endoplasmic reticulum to the Golgi apparatus, thereby affecting the activation of sterol regulatory element-binding proteins (SREBPs) [38,39]. In the bovine mammary epithelial cell line, the overexpression of SCAP and SREBP1 promotes the transfer of SREBP1 from the cytoplasm to the nucleus, as well as the expression of the *FASN* gene and lipid droplet production [40]. As a transcription factor, SREBPs play a critical role in regulating cholesterol and fatty acid levels in goat mammary glands [41]. We infer that miR-26 affects the lipid metabolism of mammary glands by targeting and downregulating the expression of INSIG1, promoting the activity of SREBPs, and mediating the synthesis of key enzymes (FASN, SCD1, and ACACA). In dairy goat mammary epithelial cells, the downregulation of *SCD1* expression could significantly reduce the concentration of C16:1 and C18:1 [42]. In knockout cells, we also detected that the expression of *SCD1* and the concentration of C18:1 were inhibited. As reported, *INSIG1* inhibits cholesterol synthesis by accelerating the degradation of 3-hydroxy-3-methylglutaryl-coenzyme A reductase (HMGR) in cells [43,44]. Taken together, we demonstrate that miR-26 mediates the target gene *INSIG1* to regulate fatty acid metabolism in goat mammary epithelial cells (Figure 8).

## 4. Materials and Methods

### 4.1. Construction of Dual-sgRNA-Cas9 Expression Vector 

The sgRNAs targeting the genomic sequence of goat pre-miR-26a and pre-miR-26b were designed using the CHOPCHOP website (http://chopchop.cbu.uib.no/, accessed on 9 November 2020). Three sgRNAs with high prediction scores and targeting both ends of the pre-miR-26a and pre-miR-26b sequences were selected, respectively (Figure 1A,B), and synthesized by Sangon Biotech (Shanghai, China) Co., Ltd. Next, the double-stranded oligonucleotides formed by the annealing of sgRNA were inserted into the Bbs *I* site of plasmid PX459 to construct a vector co-expressing sgRNA and Cas9, named 26a-sg1-PX459, 26a-sg2-PX459, 26a-sg3-PX459, 26b-sg1-PX459, 26b-sg2-PX459, and 26b-sg3-PX459, respectively. U6-26a-sg2-tracRNA, U6-26a-sg3-tracRNA, U6-26b-sg2-tracRNA, and U6-26b-sg3-tracRNA sequences were cloned from 26a-sg2-PX459, 26a-sg3-PX459, 26b-sg2-PX459, and 26b-sg3-PX459, respectively, using the PrimeSTAR Kit (Takara, Japan). The forward and reverse primers (F: 5′-CACCTCTAGAGAGGGCCTATTTCCCATGATTCCTTCATAT-3′, R: 5′-CACCGGTACCAAAAAAGCACCGACTCGGTGCCACTTTTTC-3′) were synthesized by Sangon Biotech (Shanghai). U6-26a-sg2-tracRNA and U6-26a-sg3-tracRNA sequences were inserted into 26a-sg1-PX459 by double endonucleases (Xba *I* and Kpn *I*) to construct a dual−sgRNA vector, named 26a-sg1-sg2-PX459 and 26a-sg1-sg3-PX459 (Figure 1C). 26b-sg1-sg2-PX459 and 26b-sg1-sg3-PX459 were obtained via the same method.

### 4.2. Cell Culture and Treatment

We isolated and cultured GMECs according to the previous method [20]. To induce lactation, 2 μg/mL prolactin (Sigma, St. Louis, MO, USA) was used for two days prior to the subsequent experiments [21]. To obtain GMECs with the knockout of miR-26a, when the density of GMECs reached 80% in a 6-well plate, the cells were transfected with 1 μg each of 26a-sg1-sg2-PX459 and 26a-sg1-sg3-PX459 via Lipofectamine 2000 (Invitrogen, Waltham, MA, USA). In the miR-26b knockout group, 26b-sg1-sg2-PX459 and 26b-sg1-sg3-PX459 were transfected into cells using the same method. In the miR-26ab knockout group, double sgRNA plasmids for miR-26a and miR-26b (26a-sg1-sg3-PX459 and 26b−sg1-sg2-PX459) were transfected into GMECs. The PX459 vector was a negative control. After 48 h, cells were cultured in a medium containing 1 μg/mL puromycin (Sigma, St. Louis, MO, USA) for 4 days and then changed to a normal medium (Figure 1D). For the *INSIG1* interfering assay, when the cell density reached 70–80%, the siRNA of *INSIG1* (GenePharma, Shanghai, China) was transfected with LipofectamineRNAiMax (Invitrogen, Waltham, MA, USA) and the cells were collected after 48 h.

All experimental procedures were approved by the Institutional Animal Care and Use Committee of Northwest A&F University, YangLing, China (protocol number 15-516).

### 4.3. Cell Selection and Single-Cell Clones

After 4 days of puromycin selection, GMECs with transfected vectors were cultured in a normal medium. The basal medium contained 90% DMEM/F12 (Hyclone, Logan, UT, USA), 5 μg/mL bovine insulin (Sigma, St. Louis, MO, USA), 100 U/mL penicillin/streptomycin, 10 ng/mL epidermal growth factor (Invitrogen, Waltham, MA, USA), 5 μg/mL hydrocortisone (Sigma, St. Louis, MO, USA), and 10% fetal bovine serum (Invitrogen, Waltham, MA, USA) [21]. Surviving cells were expanded. Upon reaching 80% density, the cells were collected, diluted to 1 cell per 100 μL, and cultured in 96 wells for 8–14 days. These cells formed single-cell clones that were cultured and expanded (Figure 1D).

### 4.4. DNA Extraction, PCR, and T7EN1 Cleavage

When single-cell clones filled the 12-well plate, the cells were harvested for DNA extraction using a Universal Genomic DNA Kit (CW Biotech, Beijing, China) and continued expansion. We then amplified a genomic fragment spanning pre-miR-26a and pre-miR-26b using PCR−26a and PCR−26b primer (Appendix A) and verified the modification of the genomic sequence using Sanger sequencing. Then, we used the Cas-OFFinder (http://www.rgenome.net/casoffinder/, accessed on 9 November 2020) to predict the off-target sites of 3 sgRNAs and amplified those containing off-target sites. PCR products of genomic sequences were used for off-target effect analysis via the T7EN1(T7 Endonuclease I, NEB, Ipswich, MA, USA)-cleavage assay. PCR sequence primers are displayed in Appendix A.

### 4.5. Total RNA Extraction and Quantitative Real-Time PCR (qPCR)

Total RNA was isolated from GMECs using RNAiso Plus (Takara, Kusatsu City, Japan). The quality and concentration of total RNA were determined using a microspectrophotometer (NanoDrop 2000, Thermo Fisher, Waltham, MA, USA). The ratio of optical density at 260 nm:280 nm was between 1.9 and 2.1 for all samples. The integrity of total RNA was evaluated via agarose gel electrophoresis analysis of 28s and 18SrRNA subunits. For the mature miR-26a−5p, miR-26a-3p, miR-26b-5p, and miR-26b-3p expression levels detected, 0.5 μg RNA was reverse transcripted into cDNA using the miRcute cDNA First-Strand Kit (Tiangen, Beijing, China) through poly-A, and cDNA was quantified for real-time fluorescence using the miRcute miRNA q-PCR Kit (Tiangen, Beijing, China) through SYBRGreen. The miRNAs’ relative expression levels were calculated via snRNA U6 as the internal reference gene and 2^−ΔΔCt^. The mRNA level was detected via reverse transcription of 0.5 μg total RNA using the PrimeScript RT Kit (Takara, Japan). Then, real-time fluorescence quantification was performed using SYBR TaqII (Takara, Japan). The calculation method was also 2^−ΔΔCt^, and it was normalized by comparison with ubiquitously expressed transcript (*UXT*) levels. All qPCR primers are shown in Appendix A [41,45].

### 4.6. Western Blot Analysis

For the expression level of the protein assay, cells were lysed using pre-chilled RIPA buffer (Solarbio, Beijing, China) containing protease inhibitors (Roche). The lysis product was centrifuged at 12,000× *g* for 5 min, and then the supernatant was collected as a total protein solution. Then, the concentration was measured using the BCA Protein Kit (Thermo Fisher, Waltham, MA, USA). A total of 15 μg of protein was separated by 10% SDS/PAGE and transferred onto PVDF membranes. Then, membranes were incubated at 4 °C for 10 h with the primary antibodies to the protein of interest, rabbit anti-INSIG1 (Bioss, 1:500) or mouse anti-β-actin (CWBIO, 1:1000), and washed 3 times for 10 min each time with TBST (Tris Buffered Saline with Tween 20); next, they were incubated with the secondary antibody of the species that corresponded to the primary antibody: polyclonal anti-rabbit or monoclonal anti-mouse horseradish-conjugated IgG (CW Biotech, 1:2000). After washing (3 × 10 min) with TBST, the membranes were incubated with a solution in the enhanced chemiluminescent (ECL, Bio-Rad, Hercules, CA, USA) for 1 min, and then the luminescence signal intensity was detected using the Western blotting detection system (Bio-Rad). 

### 4.7. Lipid Droplet Staining

For the lipid droplets assay, GMECs were seeded in 12-well plates and cells were washed 3 times with pre-chilled phosphate buffer solution (PBS) once the cells had grown to 60–80% confluency. GMECs were treated with 4% paraformaldehyde for 30 min at 4 °C, washed 3 times with PBS, and stained via incubation with 1 μg/mL of Bodipy staining solution to protect from light for 30 min at room temperature. Then, GMECs were washed three times with PBS, stained via incubation with DAPI staining solution in the dark for 10 min at room temperature, and next washed 3 times with PBS. The fluorescence was observed and photographed using a cell imaging detector to analyze the relative content of lipid droplets in the cells.

### 4.8. Triacylglycerol (TAG), Cholesterol, and Fatty Acid Analysis

GMECs were washed with pre-chilled PBS buffer 3 times, and the contents of cholesterol and TAG were detected via the enzymatic colorimetric method, using the Cholesterol Kit (Applygen, Beijing, China) and Triacylglycerol Kit (Applygen, China). Cholesterol and TAG concentrations were normalized to the total cell protein measured using the BCA Protein Kit (Thermo Fisher, Waltham, MA, USA). For the fatty acid assay, we extracted intracellular fatty acids from 60 mm dishes with 2 mL of sulfuric acid/methanol (2.5:100, *v*/*v*). After ultrasonication for 10 min, the lytic product was incubated at 80 °C for 1 h. After cooling to room temperature, 2 mL of 0.1 M HCl and 800 μL of hexyl hydride (Sigma, St. Louis, MO, USA) were added and the tube was vortexed for 30 s followed by centrifugation for 5 min at 900× *g*. The supernatant was collected in a 2 mL silicified tube with 0.5 g of water-free sodium sulfate. After vortexing, the tube was centrifuged for 3 min at 13,800× *g*, and the liquid supernatant was collected for fatty acid analysis using a GC-MS (GC-MS, Agilent, Santa Clara, CA, USA) instrument, and the ratio of the peak area of the fatty acid to the total peak area was considered as the relative proportion of the fatty acid [41].

### 4.9. Statistical Analysis

The statistical significance of the data was analyzed in SPSS 19. Differences between the two treatments were tested for significance using Student’s *t*-test, where differences were considered to be statistically significant when *p* < 0.05 (* *p* < 0.05, ** *p* < 0.01). One-way ANOVA was used for comparison among multiple groups, and significant differences were considered when the *p*-value was less than 0.05 (lowercase letters, *p* < 0.05).

## 5. Conclusions

In summary, we found that miR-26a and miR-26b have promoting effects on the content of lipid droplets, triglycerides, and cholesterol, as well as the expression of related genes in GMECs. Interestingly, the simultaneous deficiency of miR-26a and miR-26b increased the inhibitory effect on the synthesis of unsaturated fatty acids in GMECs, but there was no additive effect. This provides reference data for a new method to improve the milk fat content of dairy animals by editing the miRNA family in genes.

## Figures and Tables

**Figure 1 ijms-24-10028-f001:**
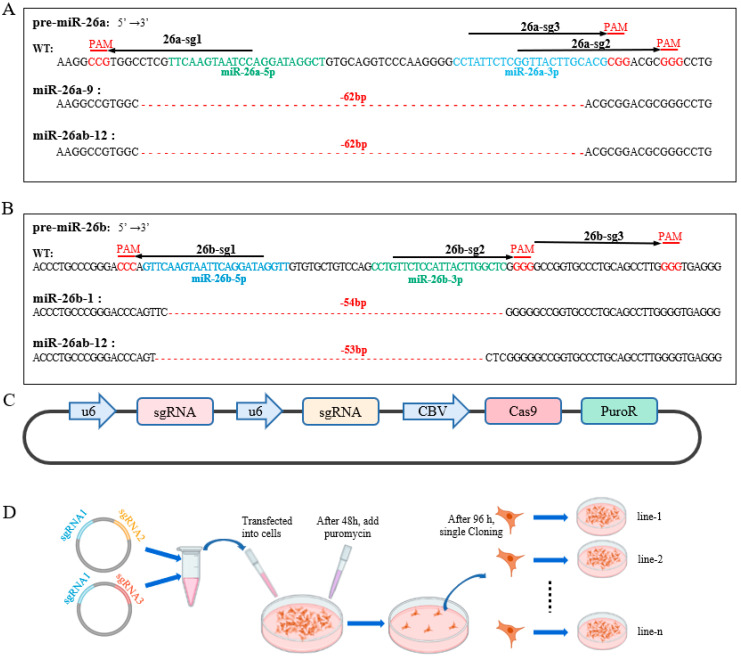
The multi-sgRNA CRISPR/Cas9 system knockout pattern of miR-26a/b. (**A**) sgRNA selection of goat miR-26a and the pre-miR-26a sequences in knockout cells. Pre-miR-26a, the precursor of goat miR-26a. Pre-miR-26b, the precursor of goat miR-26b. 26a-sg1, sgRNA-1 of pre-miR-26a. 26a-sg2, sgRNA-2 of pre-miR-26a. 26a-sg3, sgRNA-3 of pre-miR-26a. WT, wild-type GMECs. miR-26a-9, miR-26a knockout GMECs. miR-26ab-12, miR-26a, and miR-26b knockout GMECs. (**B**) sgRNA selection of goat miR-26b and the pre-miR-26b sequences in knockout cells. 26b-sg1, sgRNA-1 of pre-miR-26b. 26b-sg2, sgRNA-2 of pre-miR-26b. 26b-sg3, sgRNA-3 of pre-miR-26b. WT, wild-type GMECs. miR-26b-1, miR-26b knockout GMECs. miR-26ab-12, miR-26a and miR-26b knockout GMECs. (**C**) Double sgRNAs and CRISPR/Cas9 expression vector mode diagram. U6, U6 promoter. CBV, CBV promoter. PuroR, puromycin resistance gene. (**D**) Pattern diagram of approach to obtain miR-26a/b knockout cell.

**Figure 2 ijms-24-10028-f002:**
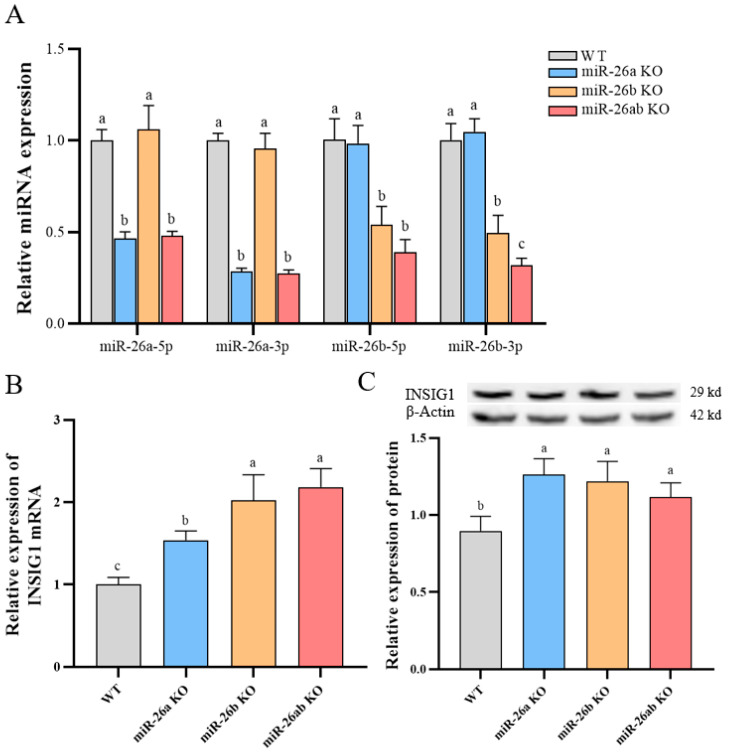
Expression of miR-26a/b and target gene INSIG1 in knockout GMECs. (**A**) miR-26a/b expression in knockout GMECs. (**B**) mRNA expression of INSIG1 in knockout GMECs. (**C**) INSIG1 protein expression in knockout GMECs. WT, wild-type GMECs. miR-26a KO, miR-26a knockout GMECs. miR-26b KO, miR-26b knockout GMECs. miR-26ab KO, GMECs with co-knockout of miR-26a and miR-26b. Data are shown as mean ± SD for three independent experiments. Lower case letters, *p* < 0.05.

**Figure 3 ijms-24-10028-f003:**
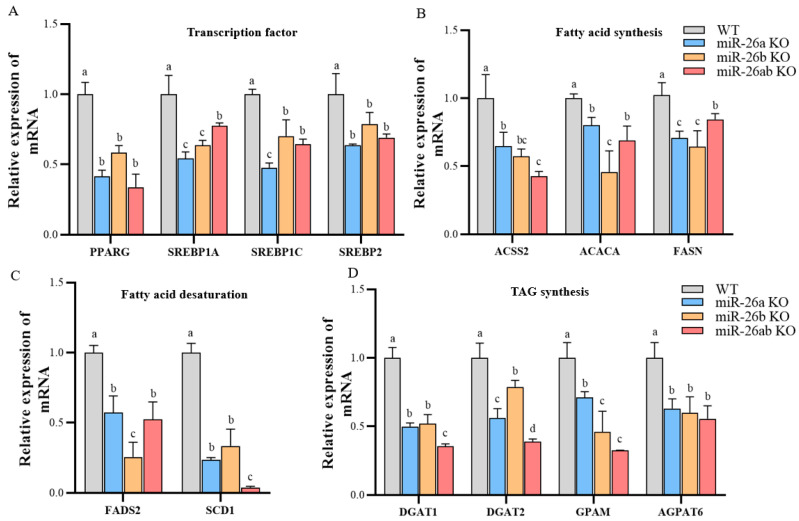
Expression of genes related to lipid metabolism in knockout GMECs. (**A**) Expression of transcription factor genes. (**B**) Expression of genes related to fatty acid synthesis. (**C**) Expression of genes related to fatty acid desaturation. (**D**) Expression of genes for TAG synthesis. WT, wild-type GMECs. miR-26a KO, miR-26a knockout GMECs. miR-26b KO, miR-26b knockout GMECs. miR-26ab KO, GMECs with co-knockout of miR-26a and miR-26b. Data are shown as mean ± SD for three independent experiments. Lower case letters, *p* < 0.05.

**Figure 4 ijms-24-10028-f004:**
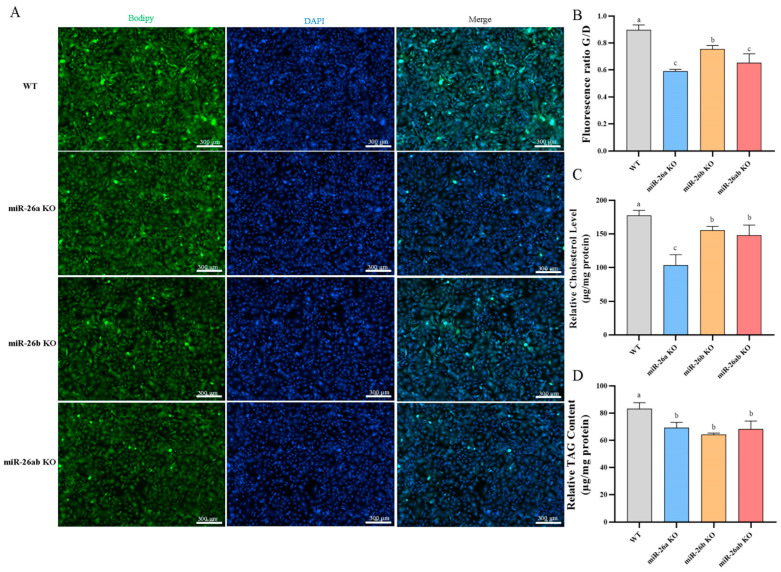
The contents of lipid droplets, cholesterol, and TAG in knockout GMECs. (**A**,**B**) The contents of lipid droplets. Bar, 300 μm. (**C**) The contents of cholesterol. (**D**) The contents of TAG. WT, wild-type GMECs. miR-26a KO, miR-26a knockout GMECs. miR-26b KO, miR-26b knockout GMECs. miR-26ab KO, GMECs with co-knockout of miR-26a and miR-26b. Data are shown as mean ± SD for three independent experiments. Lower case letters, *p* < 0.05.

**Figure 5 ijms-24-10028-f005:**
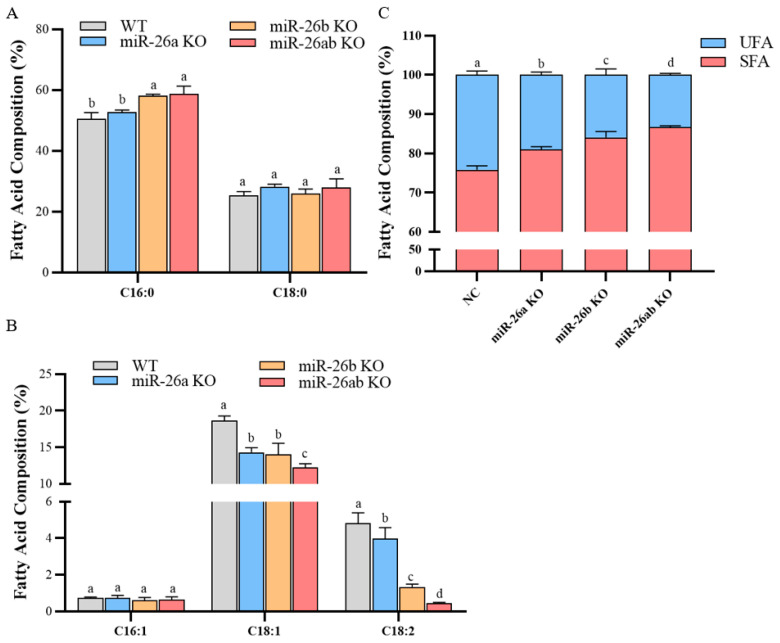
The percentage of fatty acids in knockout GMECs. (**A**) The percentage of saturated fatty acids (C16:0 and C18:0). (**B**) The percentage of unsaturated fatty acids (C16:1, C18:1, and C18:2). (**C**) The percentage of total saturated fatty acids (SFAs) and unsaturated fatty acids (UFAs). C16:0, palmitic acid. C18:0, stearic acid. C16:1, palmitoleic acid. C18:1, oleic acid. C18:2, linoleic acid. WT, wild-type GMECs. miR-26a KO, miR-26a knockout GMECs. miR-26b KO, miR-26b knockout GMECs. miR-26ab KO, GMECs with co-knockout of miR-26a and miR-26b. Data are shown as mean ± SD for three independent experiments. Lower case letters, *p* < 0.05.

**Figure 6 ijms-24-10028-f006:**
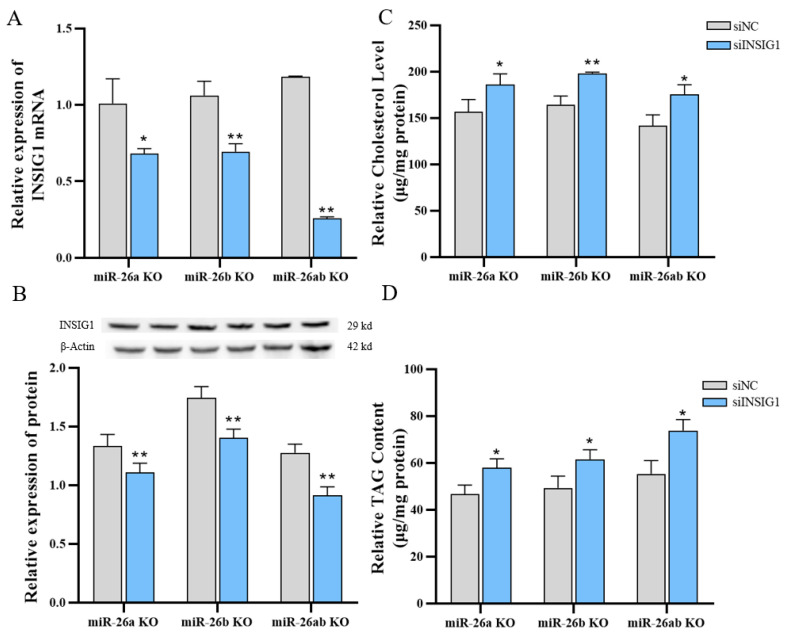
The contents of cholesterol and TAG in knockout GMECs with inhibiting INSIG1. (**A**) Expression of *INSIG1* mRNA in siINSIG1-treated knockout GMECs. (**B**) INSIG1 protein expression in siINSIG1-treated knockout GMECs. (**C**) The contents of cholesterol in knockout GMECs with silent *INSIG1*. (**D**) Triglyceride content in knockout GMECs with silent *INSIG1*. siNC, negative control of interfering RNA. siINSIG1, interfering RNA of *INSIG1*. miR-26a KO, miR-26a knockout GMECs. miR-26b KO, miR-26b knockout GMECs. miR-26ab KO, GMECs with co-knockout of miR-26a and miR-26b. Data are shown as mean ± SD for three independent experiments. *, *p* < 0.05. **, *p* < 0.01.

**Figure 7 ijms-24-10028-f007:**
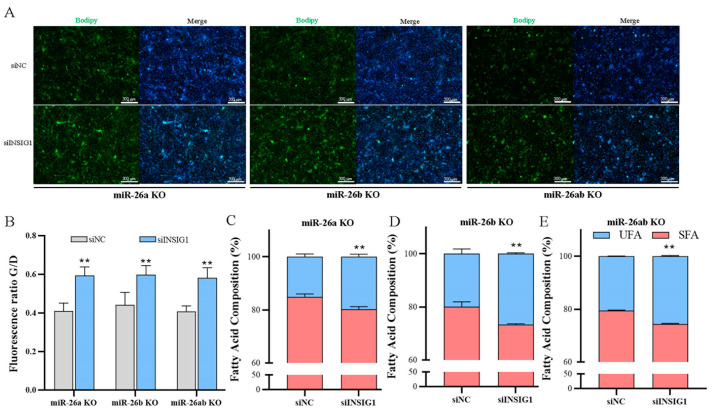
The contents of lipid droplets and the percentage of fatty acid in knockout GMECs with silent *INSIG1*. (**A**,**B**) The contents of lipid droplets in knockout GMECs with inhibiting *INSIG1*. Bar, 300 μm. (**C**–**E**) The percentage of unsaturated fatty acids in knockout GMECs with silent *INSIG1*. siNC, negative control of interfering RNA. siINSIG1, interfering RNA of *INSIG1*. miR-26a KO, miR-26a knockout GMECs. miR-26b KO, miR-26b knockout GMECs. miR-26ab KO, GMECs with co-knockout of miR-26a and miR-26b. Data are shown as mean ± SD for three independent experiments. **, *p* < 0.01.

**Figure 8 ijms-24-10028-f008:**
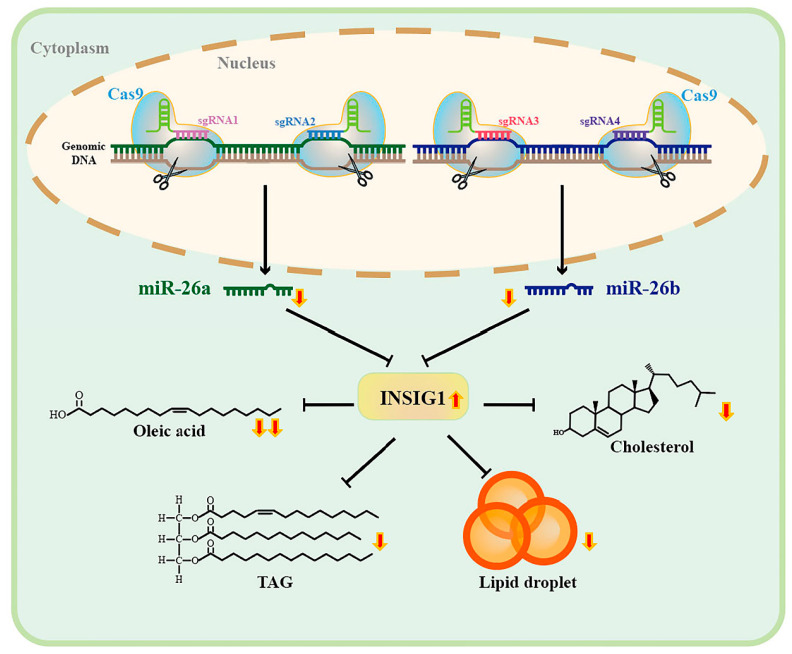
Lipid metabolism is suppressed in miR-26a/b knockout goat mammary epithelial cells by upregulating *INSIG1*.

## Data Availability

The original contributions presented in the study are included in the article/Appendix A; further inquiries can be directed to the corresponding authors.

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
