# Peer review of "Fatty Acid Desaturation Is Suppressed in Mir-26a/b Knockout Goat Mammary Epithelial Cells by Upregulating INSIG1"

_ijms, 2023, doi:10.3390/ijms241210028_

Round 1

Reviewer 1 Report

The two micro-RNA 26 (miR26a and miR26b) were knock out with CRISPR/cas9 in goat mammary epithelial cells (GMEC), in order, to get insight on their effects on the fatty acid metabolism. Knock out cells showed a decrease in fatty acid metabolites and increase in the expression of mRNA INSIG1 genes. Silencing the INSIG1 gene in the mir26 KO cells restored the level of fatty metabolites in these cells, suggesting a role for INSIG1 in the mir26 effects on fatty acid metabolites. 

The paper writing gives the impression of being a first draft. The results would benefit from being better highlighted and discussed compared to the previous paper by Wang et al, (2016) who had worked on the same topic.  In the version presented here, it seems that the interest was technical (CRISPR/Cas) rather than novelties in the results.

The English writing should be improved.

The material and methods should be more consistent, some parts are well described and other less. It is important that the minimal requirements for the qPCR analysis are described, see article: http://www.clinchem.org/cgi/doi/10.1373/clinchem.2008.112797. Care should also be given to the description of the protein and lipid quantifications since they are the results bases. 

Supplementary tables should be edited to present the same way the data and PCR efficiency for each primer pairs should be shown. 

It appears that there are 3 miR26 loci (Wang et al., 2016), why are only two presented here? 

The authors did not demonstrate a direct effect of miR26 KO on INSIG1 (as stated in the abstract) but showed that downregulation of INSIG1 counteract the miR26 KO effect on lipid biosynthesis. This was previously showed in Wang 2016, 2108.

Minor comments

Abstract

L11 and 12: the two sentences can be merged.

Introduction

L26-34 should be removed.

L58-60: The sentence: “The genes expression….” misses a reference. 

L64-69: maybe this paragraph is not useful since it is known that CRISPR/Cas system works.

Results

L80: No need of “respectively”.

L78-87: this part is very interesting and informative and belongs to the methods with Figure 1.

L89: What are the “miR-26a-5p and miR-26a-3p”.

L126: “miR-b” is miR-26b?

L122-128: this paragraph should be in the methods.

L128: the two host genes must be introduced before.

L139-151: the genes used in this study should be presented in the introduction.

L176: “percentag” should be percentage

Discussion

L237-240: remove

L250-267: this paragraph should be presented in the results.

Materials and Methods

L346: which sequences?

L367: it looks like the same constructs were used for miR26a and 26b.

L395: “T7EN1 » should be explained

L399 : « Takara » from where to be consistent.

L399: what is the obtained quality of total RNA?

L403: The “Minute kit” does it work with specific primers or random primers or polyT for cDNA synthesis and for qPCR, is it SybrGreen- or on probes-based?

L406: what are the efficiencies of each qPCRs? 

L406: why a second kit for mRNAs?

L413: “Total protein was supernatant obtained by centrifugation at 12,000 g for 5 min.” Do the authors mean, that the protein fraction was collected in the supernatant after centrifugation at 12,000 g for 5 min?

L418: the authors should describe what they used for the washing step.

L421: what is the “chromogenic solution”?

L439: The fatty acid method should be better described with concentration of the reagents or at least how many cells in the extracts for how many ml of fixative.

The English should be improved. Some phrases sound unclear.

Reviewer 2 Report

In the introduction lines 26-34 are instructions they should be taken out.

In the discussion line 237-240 are also the instructions.

The study is good and it revisits the known association between the miR26 and INSIG1.

The study should have included the add back of the host gene CTDSP, as is done for siRNA mediated knock down of the miR Target.

The effect of the CTDSP getting affected by the miR26 K/O will have its impact on biology and bringing back the product of this gene will bring some incites on just the impact of the miR 26 ko.

The manuscript needs to revisited for its discussion section and discussing the contribution of the different biosynthetic enzymes that are assayed and how do they connect to the whole process of fatty acid biosynthesis.

A model picture connecting all the players in the study and relying to the biological process will be a benefit to the readers.

Language is Ok.
